# Immune Checkpoint Inhibitor Rechallenge in Renal Cell Carcinoma: Current Evidence and Future Directions

**DOI:** 10.3390/cancers15123172

**Published:** 2023-06-13

**Authors:** Enrico Sammarco, Fiorella Manfredi, Amedeo Nuzzo, Marco Ferrari, Adele Bonato, Alessia Salfi, Debora Serafin, Luca Zatteri, Andrea Antonuzzo, Luca Galli

**Affiliations:** 1Unit of Medical Oncology 2, Azienda Ospedaliero-Universitaria Pisana, Santa Chiara Hospital, 56126 Pisa, Italy; enricosammarco1992@gmail.com (E.S.); manfredifiorella@gmail.com (F.M.); amedeonuzzo@gmail.com (A.N.); marcoferrari0201@gmail.com (M.F.); adelebonato@gmail.com (A.B.); alessiasalfi@gmail.com (A.S.); serafindebora@gmail.com (D.S.); luca3zatteri@gmail.com (L.Z.); 2Unit of Medical Oncology 1, Azienda Ospedaliero-Universitaria Pisana, Santa Chiara Hospital, 56126 Pisa, Italy; automezzo69@gmail.com

**Keywords:** renal cell carcinoma (RCC), immune checkpoint inhibitors (ICIs), rechallenge, resistance, combination treatment

## Abstract

**Simple Summary:**

The advent of immune checkpoint inhibitors has dramatically changed the history of advanced renal cell carcinoma treatment. Their use in first-line therapy provides an undeniable advantage in terms of survival; nevertheless, a considerable proportion of patients undergo disease progression. Similarly to other advanced solid tumors, even in renal cell carcinoma, preliminary data are beginning to emerge concerning the activity and the efficacy of a rechallenge of an immune checkpoint inhibitor-based treatment. In this mini-review, we summarize available data about immunotherapy rechallenge in renal carcinoma and its future perspectives.

**Abstract:**

Immune checkpoint inhibitor-based therapies represent the current standard of care in the first-line treatment of advanced renal cell carcinoma. Despite a clear benefit in survival outcomes, a considerable proportion of patients experience disease progression; prospective data about second-line therapy after first-line treatment with immune checkpoint inhibitors are limited to small phase II studies. As with other solid tumors (such as melanoma and non-small cell lung cancer), preliminary data about the clinical efficacy of rechallenge of immunotherapy (alone or in combination with other drugs) in renal cell carcinoma are beginning to emerge. Nevertheless, the role of rechallenge in immunotherapy in this setting of disease remains unclear and cannot be considered a standard of care; currently some randomized trials are exploring this approach in patients with metastatic renal cell carcinoma. The aim of our review is to summarize main evidence available in the literature concerning immunotherapy rechallenge in renal carcinoma, especially focusing on biological rationale of resistance to immune checkpoint inhibitors, on the published data of clinical efficacy and on future perspectives.

## 1. Introduction

The advent of immune checkpoint inhibitors (ICIs) has radically changed the treatment of advanced renal cell carcinoma (RCC), resulting in a meaningful impact on survival in these patients, with a good tolerability profile.

Nivolumab monotherapy demonstrated clinically significant improvement in overall survival compared to Everolimus in patients who had advanced RCC with a clear cell component and had previously been treated with antiangiogenic therapy [1]. More recently, randomized clinical trials have shown superiority in terms of the efficacy of immunotherapy-based combination therapies over receptor tyrosine kinase inhibitor (TKI) monotherapy in previously untreated advanced RCC. In the last few years, combination therapies based on the use of checkpoint inhibitor (CPI) and TKI, including Pembrolizumab/Axitinib [2], Pembrolizumab/Lenvatinib [3], Nivolumab/Cabozantinib [4], and Avelumab/Axitinib [5], have demonstrated longer progression-free survival than Sunitinib in patients with untreated advanced RCC; these treatments are currently approved in Europe in the first-line treatment of advanced RCC, regardless of Heng score. A further therapeutic approach, based on dual immune checkpoint inhibition with Nivolumab/Ipilimumab, has demonstrated superior long term survival benefits than Sunitinib in patients with untreated advanced RCC with intermediate/poor risk [6] and is currently available in Europe. In accordance with these results, the main international guidelines recommend the use of an immunotherapy-based combination treatment in first-line therapy of advanced RCC [7,8].

Despite these clear advantages, most patients experience disease progression, requiring the choice of a new systemic treatment. Prospective data to determine the efficacy of further treatments after an ICI-based first-line therapy are lacking and are limited to small phase II trials focused on the use of TKI [9,10,11,12]. The role of ICI rechallenge (defined as the reintroduction of an ICI-based therapy after disease progression to previous ICI therapy) in this setting remains unclear; this strategy is not currently considered a standard of care in the treatment of advanced RCC, but it may be a reasonable option according to clinical activity data from other diseases, such as advanced melanoma [13] and non-small cell lung cancer (NSCLC) [14].

In this review, we will discuss the main evidence currently available in the literature on the role of the rechallenge of immunotherapy (mainly given as combination of two different ICIs or as combination of an ICI with other types of drugs) in metastatic RCC, focusing in particular on the biological rationale of resistance to CPIs, on the reported clinical activity and safety data and on future perspectives.

## 2. Biological Rationale of Resistance to Immunotherapy

The onset of mechanisms of resistance to immune checkpoint inhibitors is linked to complex molecular alterations, which are still not completely known. The timing of occurrence makes it possible to distinguish two main kinds of resistance: primary resistance occurs in never-responder patients, while secondary resistance emerges after an initial period of tumor regression. The lack of response to immunotherapy is the result of a continuous interaction between tumor cells and the immune system. In particular, primary resistance could be explained by intrinsic tumor features, while the resistance acquired during treatment could be related to microenvironment changes.

### 2.1. Intrinsic Tumor Features

Several factors intrinsic to the tumor cell population have been involved to explain the different responses to immune checkpoint blockade.

Neoantigens are peptides that are usually absent in normal cells, but they can be produced from somatic mutations during the process of carcinogenesis or from viral peptides in some kinds of cancers (e.g., Human Papilloma Virus and cervical cancer) [15]. An increased load of neoantigens plays a key role in favoring tumor immunogenicity and is correlated with response to the immune checkpoint inhibitor (both anti-CTLA4 and anti-PD-1 therapies) through enhanced T cell response [16]. Anti-PD-1 treatment can lead to clonal evolution within the tumor population, resulting in a reduction in mutational burden and neoantigens load and, consequently, in a loss of sensitivity to immunotherapy [17].

The interferon gamma (IFN-γ) signaling pathway significantly affects the T cell response towards tumor cells: the binding to its own receptor results in the activation of Janus kinase (JAK) and signal transducer and activator of transcription proteins (STAT) [18,19], with the subsequent induction of PD-L1 expression. Inactivating mutations in the genes of the IFN-γ signaling pathway (especially JAK) protect human melanoma-derived cell lines from antitumor immune response [20].

The downregulation of MHC (major histocompatibility complex) protein expression on the surface cell may prevent the response to immune checkpoint inhibitors by reducing antigen presentation through the MHC class I and II pathway [21]. Specific alterations (deletions, point mutations, or loss of heterozygosity) in the essential components of the MHC class I, such as beta-2-microglobulin, are found more frequently in patients with melanoma unresponsive to anti-PD-1 or anti-CTLA4 treatment [22].

The WNT-β-catenin pathway is implicated in the oncogenesis process and is overexpressed in most cancers. Its hyperactivation can limit the activity of the immune system through various mechanisms, such as the production of immunosuppressive cytokines (such as interleukin 10, IL-10 [23]) or the activation of the indoleamine 2,3-dioxygenase 1 (IDO1), responsible for the development of regulatory T cells [24].

Cyclin-dependent kinase (CDK) 4 and 6 are members of a family of enzymes essential for the regulation of the cell cycle; their mutations were found to be associated with the tumorigenesis of a variety of cancers. T cell exclusion and immune evasion driven by CDK4/6 was described in melanoma samples from patients with resistance to immune checkpoint blockers [25].

Tumoral cell lines that have constitutively active mitogen-activated protein kinase (MAPK) signaling may generate an immunosuppressive environment through the synthesis of several soluble factors (such as IL-6 and IL-10) capable of reducing the activity of immune cells [26].

The loss of expression of tumor suppressor phosphatase and tensin homolog (PTEN) is a frequent event in many cancer types and promotes the proliferation and survival of tumor cells; more recently, it has been correlated with the expression of immunosuppressive cytokines, resulting in decreased T cell infiltration and T cell-mediated cell death in preclinical models [27].

### 2.2. Tumor Microenvironment

The tumor microenvironment (TME) is a complex and constantly evolving entity, composed of stromal and immune cellular elements, blood vessels, and extracellular matrix. The dynamic interaction of TME with the tumor cell population promotes its survival, local invasion, and metastatic dissemination; it also modulates the response of tumor cells to therapies.

RCC is characterized by the high density of T cells, in particular CD8+ tumor infiltrating lymphocytes (TILs), whose presence seems to be associated with worse clinical outcomes [28]. In exploratory biomarker analyses from IMmotion150 (phase II trial comparing first-line treatment in RCC with the combination of Atezolizumab with or without Bevacizumab versus standard therapy with Sunitinib), the high expression of the T-effector gene signatures was associated with the high expression of PD-L1 and CD8+ TILs and improved PFS with Atezolizumab/Bevacizumab versus Sunitinib, while no differences in comparison to Atezolizumab monotherapy were found [29].

Myeloid-derived suppressor cells (MDSCs) are regulatory cells that play a role in tumor progression by promoting immune evasion; the high frequency of circulating MDSCs may predict poor response to Ipilimumab in melanoma patients [30]. Similarly, the subgroup of patients with RCC and a high expression of myeloid inflammation gene signature had a poor response from Atezolizumab monotherapy in IMmotion150, confirming the immunosuppressive role of this cell type [29].

Tumor-associated macrophages (TAMs) represent a class of immune cells highly expressed in TME; they can be divided into two main subgroups, known as M1 and M2. While M1 TAMs have an antitumoral action, M2 macrophages can promote tumor cell proliferation and dissemination. Human RCC cells produce several factors (including IL-6 and IL-10) that induce the polarization of TAMs towards M2 phenotype. They can directly suppress T cell function and, consequently, reduce the efficacy of CPI [31].

Hypoxia, caused by the disorganized growth of tumoral vascular architecture, plays a crucial role in immune escape in RCC. Increased levels of HIF1a and HIF2a (hypoxia-inducible factors) may enhance the expression of immune checkpoints CTLA4 and PDL1 on dendritic cells and inhibit the response of T cells [32].

## 3. Literature Search and Selection of Trials

A literature search of the available data about the rechallenge of an ICI-based therapy in advanced RCC was performed by searching PubMed for all relevant publications from its inception to 1 April 2023. We included in our search the full text of articles and abstracts that were presented at main international conferences. Furthermore, we conducted a search of ongoing clinical trials by using the electronic database of Clinical Trials.

## 4. ICI Rechallenge in RCC: Current Evidence of Therapeutic Strategies for Overcoming Resistance

### 4.1. Combination of ICIs

The use of combination of drugs targeting different immune checkpoints represents a widely studied and currently used strategy in several diseases. This approach could improve the response to immunotherapy in previously untreated patients or allow a rechallenge of ICIs in patients who have experienced progression disease.

Adding a monoclonal antibody directed against CTLA4 to an anti-PD1 can bypass resistance to single agent immunotherapy; dual immune checkpoint blockade with Nivolumab/Ipilimumab has already been approved in previously untreated renal cell carcinoma and many other diseases, such as melanoma, NSCLC, and colorectal cancer. The clinical benefit of combining these two drugs derives from their complementary mechanisms: anti-CTLA4 plays a crucial role in T cell priming by enhancing their activation, whereas anti-PD1 is involved in the reversion of T cell exhaustion and subsequent reactivation of effector response [33].

#### 4.1.1. Retrospective Data

A retrospective analysis evaluated antitumor activity and safety of 45 patients with metastatic RCC who had prior exposure to anti-PD1 or anti-PDL1 antibodies and were subsequently treated with salvage Ipilimumab and Nivolumab. More than half of the patients had received at least three prior lines of treatment. Of the 45 patients, 27 (60%) received monotherapy with prior anti-PD-1 or anti-PDL-1 antibodies, 8 (18%) received an ICI that targeted the PD-1 pathway in combination with a VEGF receptor inhibitor (Axitinib, Sunitinib, or Cabozantinib), 4 (9%) received an ICI in combination with Bevacizumab, and 6 (13%) received an ICI in combination with another drug. The objective response rate (ORR) to salvage Ipilimumab/Nivolumab was 20%, whereas the disease control rate (DCR) was 36% and the median PFS was 4 months. The immune-related adverse events (irAEs) of any grade were reported by 29 (64%) of 45 patients; grade 3 irAEs were recorded in 6 (13%) patients. The most common grade 3–4 irAEs were hepatotoxicity (7%) and pneumonitis, rash, diarrhea, thrombocytopenia, and colitis (2% each) [34]. Similar results were reported by another multicenter retrospective study of 69 patients with mRCC who received rechallenge immunotherapy (as single agent or in combination with another ICI or another drug, such as TKI). The ORR was 30% in patients receiving single ICI as rechallenge immunotherapy, 25% in those receiving dual checkpoint blockade, and 23% in those receiving ICI in combination with target therapy. ICI rechallenge showed a manageable safety profile, with 16% of patients developing grade 3–4 irAEs. The risk of experiencing an irAE with CPI rechallenge was higher in patients who had an irAE with a previous line of immunotherapy (41%) compared with those who did not (20%) [35].

#### 4.1.2. Data from Prospective Trials

More recently, data from prospective trials evaluating salvage treatment with Nivolumab/Ipilimumab have been published.

FRACTION-RCC (Fast Real-time Assessment of Combination Therapies in Immuno-Oncology Study in Patients with advanced RCC) is a signal-seeking randomized phase 2 trial with an adaptive-platform design; in track 2, the efficacy and safety outcomes of treatment with Nivolumab/Ipilimumab in patients with metastatic RCC whose disease previously progressed during or after ICI were evaluated. All 46 patients included received previous anti-PD1 or anti-PDL1 therapy; half of the patients had received at least three lines of treatment previously. After a median follow up of 33.8 months, the ORR in the whole population was 17.4%, with eight partial responses and no complete responses. Stable disease was achieved as the best overall response (BOR) in 19 of 46 patients (41.3%), while 14 patients (30.4%) developed progressive disease as BOR. Antitumor activity was not evaluable or available in five patients. The median time to response was 2.9 months, with a median duration of response (DOR) of 16.4 months. Of the eight responders patients, five presented an ongoing response. Median PFS (95% CI) was 3.7 (2.0–7.3) months, while median OS (95% CI) was 23.8 (13.2 to not estimable) months. Grade 3–4 treatment-related adverse events (TRAEs) were reported in 13 patients (28.3%), TRAEs leading to discontinuation of treatment were reported in 4 patients (8.7%) [36].

In addition, several clinical trials have aimed to define a sequential use of immunotherapies, through an adaptive strategy of treatment intensification (or discontinuation) based on response.

In OMNIVORE, a multicenter, phase 2 adaptive trial, 83 patients with ICI-naïve advanced RCC were enrolled and received Nivolumab monotherapy (240 mg every 3 weeks) with subsequent arm allocation based on response within 6 months: patients who developed a complete or partial response discontinued Nivolumab and were observed (arm A), whereas patients with stable disease or progressive disease received two doses of Ipilimumab and continued Nivolumab (arm B; in combination treatment, patients received Nivolumab 3 mg/kg with Ipilimumab 1 mg/kg every 3 weeks). The primary endpoint in arm B was the proportion of Nivolumab non-responders who were converted to responders after the addition of Ipilimumab; only 2 of 57 patients (4%) allocated to arm B experienced a conversion to a confirmed partial response, with no complete response observed [37].

An adaptive strategy was assessed in another phase 2 trial, the cohort A of HCRN GU16-260. Eligible patients with previously untreated advanced RCC received Nivolumab monotherapy as first-line treatment until progressive disease, toxicity, or completing 96 weeks (part A); subsequently, patients experiencing progressive disease before or stable disease at 48 weeks could receive salvage treatment with Nivolumab/Ipilimumab (part B). Of 123 enrolled patients, 35 received salvage treatment in part B; the ORR by RECIST to Nivolumab/Ipilimumab was 4 of 35 (11.4%) with 1 complete response, whereas by irRECIST was slightly greater (17.2%) [38].

TITAN-RCC (Tailored ImmunoTherapy Approach with Nivolumab in RCC), a multicenter, phase 2 trial, evaluated the activity and safety of a tailored immunotherapy approach in patients with intermediate/poor risk (by IMDC score) and CPI-naive advanced RCC with a clear cell component. Of 209 recruited patients, 109 were previously untreated, and 98 received a first-line treatment with TKI for RCC. All patients received Nivolumab monotherapy; early progressors (at week 8) or non-responders (at week 16) received 2–4 doses of Ipilimumab in combination with Nivolumab, while responders to Nivolumab monotherapy continued with maintenance and could receive Nivolumab/Ipilimumab in case of subsequent disease progression. Of all patients, 67% (139/207) received at least one boost cycle of Ipilimumab; the rescue strategy with dual checkpoint blockade led to ORR of 17% in both patients who previously received Nivolumab monotherapy as first-line and second-line treatment [39,40].

The activity, efficacy, and safety of salvage treatment with Nivolumab/Ipilimumab were also reported by a recent meta-analysis that included (three out of the seven trials had an adaptive design) 310 ICI-pretreated patients from a total of seven studies (three out of the seven trials had an adaptive design). The ORR to dual checkpoint blockade was 14% higher in the standard trials compared to adaptive ones (21% vs. 10%, respectively). There was no correlation between response to prior immunotherapy and response to salvage dual checkpoint inhibition. Median PFS ranged between 3.7 and 5.5 months, while the overall incidence of grade 3–4 AEs was 27% [41].

### 4.2. Combination of Antiangiogenics Drugs and ICIs

The role of drugs targeting the vascular endothelial growth factor receptor (VEGFR) and its pathway in improving the response to ICI is well known and based on a strong rationale. Antiangiogenic drugs determine the normalization of the tumor vascular structure, consequently increasing immune cell infiltration [42]. In addition, these drugs may restore an immune-sensitive TME by reducing the levels of cells with immunosuppressive functions, such as MDSCs and regulatory T cells and promoting the differentiation of monocytes into mature dendritic cells [43]. Hypoxia alleviation afforded by TKIs also limits the differentiation of macrophages toward the M2 phenotype, endowed with immunosuppressive activity [44].

#### Data from Prospective Trials

In IMmotion150, a multicenter, open-label, phase 2 trial, 305 patients with untreated metastatic RCC (with clear cell component and/or sarcomatoid component) were randomized to receive Atezolizumab/Bevacizumab, Atezolizumab, or Sunitinib. After disease progression on Atezolizumab or Sunitinib, cross-over to Atezolizumab/Bevacizumab was allowed. Overall, 44 patients in the first-line Atezolizumab arm received Atezolizumab/Bevacizumab after disease progression within the second-line part of the trial. The ORR to second-line treatment was 25%; 4 patients of 21 (19%) with progressive disease, as the best response to first-line Atezolizumab monotherapy, achieved a partial response on Atezolizumab/Bevacizumab; the median PFS was 11.1 months [45].

Data on the activity of combination of antiangiogenics and ICIs on a larger sample size are provided by KEYNOTE-146, an open-label, single-arm, phase 1b/2 study of Lenvatinib/Pembrolizumab in patients with selected advanced solid tumors. Overall, 145 patients with metastatic RCC (with predominant clear cell component) were enrolled in this trial; three groups were identified: treatment-naïve patients, patients previously treated without ICI, and patients previously treated with at least one prior CPI (anti-PD1 or anti-PDL1 therapy). The latter group represented the majority of the whole population (104 out of 145); almost all of the ICI-pretreated patients (92.3%) had an ICI as their most recent treatment. Nivolumab/Ipilimumab was the previous ICI-based therapy in 39 patients, while 18 patients received anti-VEGF therapy in combination with ICI and 47 patients received an ICI with or without other treatment. The ORR by investigator assessment in the whole ICI-pretreated population was 62.5%, with a median duration of response of 12.5 months; in the Nivolumab/Ipilimumab subgroup, the ORR was higher (61.5%) compared to anti-VEGF/ICI subgroup (38.9%) and was similar to those who received ICI with or without other therapy (57.4%). Median PFS was 12.2 months, while median OS was not reached (median follow up time of 16.6 months). All patients developed at least one TRAE, while grade 3–5 TRAEs were reported in 64% of patients. There were two treatment-related deaths (gastrointestinal hemorrhage and another not otherwise specified death); hypertension was the most common grade 3 TRAE (21% of patients) [46].

Currently available data (on 1 April 2023) from prospective trials concerning ICI rechallenge are reported in Table 1.

## 5. Ongoing Trials of ICI Rechallenge in RCC

### 5.1. SNAPI

Sitravatinib is a novel TKI that targets TAM receptors (such as TYRO3 and AXL) and VEGFR2, which appear to be implicated in TME immunomodulation [47]. The use of Sitravatinib in combination with ICI could lead to a reduction in MDSCs and the repolarization of macrophages toward the M1 type, resulting in a less immunosuppressive TME [48]. Sitravatinib/Nivolumab showed encouraging results in terms of antitumor activity and OS in patients with non-squamous NSCLC who progressed on prior ICI [14].

SNAPI (Sitravatinib and Nivolumab After Prior Immunotherapy) is an open-label, phase 2 trial, whose aim is to explore activity, efficacy, and tolerability of this combination therapy in advanced RCC (with clear cell component). Eligible patients must have received 1 or 2 prior lines of systemic therapy for metastatic RCC, and the most recent treatment must include anti-PD1 or anti-PDL1. Investigators plan to enroll 88 participants; the primary endpoints are ORR and DCR at 24 weeks, secondary endpoints are OS, PFS, DOR, 1-year OS, safety, tolerability, and impact on health-related quality of life (NCT04904302).

### 5.2. TiNivo-2

Tivozanib is a potent and selective tyrosine kinase inhibitor of VEGFR1, 2, and 3 with a long half-life (approximately 4 days) [49]. In TIVO-1, a randomized, open-label, phase 3 trial, Tivozanib showed a significative PFS advantage compared with Sorafenib as initial VEGF-targeted therapy [50] and its use was approved in the European Union as a first-line treatment of advanced RCC and for patients who are VEGFR and mTOR pathway inhibitor-naïve following disease progression after one prior treatment with cytokine therapy for advanced RCC [51]. Afterwards, Tivozanib was compared with Sorafenib in a heavily pretreated RCC population (as third line or forth line therapy) in TIVO-3, again showing a PFS improvement. Interestingly, about a quarter of the included patients had received prior ICI therapy. Tivozanib showed a significant benefit in terms of PFS in this subgroup as well [52]. Preliminary data about the safety profile and antitumor activity of Tivozanib/Nivolumab combination treatment in metastatic RCC have been published [53].

TiNivo-2 is an open-label, randomized, multicenter phase 3 trial that compares Tivozanib monotherapy with Tivozanib/Nivolumab. Approximately 326 patients with previously treated advanced clear cell RCC will be randomized in a 1:1 ratio to Tivozanib monotherapy or Tivozanib/Nivolumab. To be eligible, patients must have received 1 or 2 prior lines of treatment and have experienced disease progression during or following at least 6 weeks of treatment with ICI; patients will be stratified by IMDC risk score and whether immunotherapy was received in most recent line or not. The primary endpoint is PFS, while secondary endpoints are OS, ORR, DOR, and safety (NCT04987203) [54].

### 5.3. CONTACT-03

Cabozantinib is a small-molecule inhibitor of several tyrosine kinases, including MET, VEGFR, and AXL [55]. In METEOR, a phase 3 trial, Cabozantinib treatment was associated with PFS and OS improvement compared with Everolimus in patients with advanced RCC, who progressed after previous VEGFR TKI [56,57], and was approved for this population. Afterwards, it was approved as a first-line treatment in intermediate/poor-risk patients based on the results of the randomized phase 2 CABOSUN [58]. More recently, the combination Cabozantinib with Nivolumab was approved in previously untreated advanced RCC patients due to the PFS and OS advantage demonstrated by the phase 3 trial CheckMate 9ER [4,59]. In COSMIC-021, a phase 1b trial, Cabozantinib/Atezolizumab showed encouraging clinical activity and acceptable safety profiles in patients with advanced RCC [60].

The open-label, randomized, phase 3 study CONTACT-03 is evaluating Cabozantinib with Atezolizumab compared with Cabozantinib monotherapy as second-line or third-line treatment in advanced RCC (with or without clear cell component). Eligible patients must have received CPI-based treatment immediately preceding the line of therapy. This trial plans to randomize 523 subjects in a 1:1 ratio to Cabozantinib/Atezolizumab or Cabozantinib, stratifying them by IMDC risk score, histology, and line of most recent prior CPI treatment. The primary endpoints are OS and PFS by the Independent Review Facility (IRF); secondary endpoints are investigator-assessed PFS and IRF- and investigator-assessed ORR and DOR, while safety, health-related quality of life and biomarker analysis are additional endpoints (NCT04338269) [61].

### 5.4. Entinostat in Combination with Nivolumab/Ipilimumab

Histone deacetylase (HDAC) inhibitors are a class of new cytostatic drugs that inhibit tumor cell proliferation by targeting enzymes implicated in the regulation of chromatin structure and in the removal of acetyl groups from histone tails [62]. Preclinical studies have reported a synergic activity between HDAC inhibitors and immunotherapy linked to an increase in effector T lymphocytes [63]. Entinostat is a selective oral inhibitor of class 1 HDAC with a half-life of 140 h [64]; in a phase 1 study, Entinostat, in combination with Nivolumab with or without Ipilimumab, demonstrated a manageable safety profile and the preliminary evidence of clinical efficacy in different advanced solid tumors [65].

HCRN GU17-326 is a single-arm, phase 2 trial that explores the activity and safety of Entinostat in combination with Nivolumab/Ipilimumab in patients with advanced RCC who have progressed on the Nivolumab/Ipilimumab regimen. In a previous safety-lead-in phase, investigators will have established the recommended phase 2 dose (RP2D) of Entinostat when used in combination with Nivolumab/Ipilimumab. The primary endpoint is ORR, secondary endpoints are safety, ORR by irRC (immune related response criteria), PFS, PFS by irRC, and OS (NCT03552380).

A brief summary of the main features of selected ongoing trials is reported in Table 2.

## 6. Conclusions

The main purpose of our work is to offer a concise overview of the available data and the future prospects inherent in the ICI-based treatment rechallenge in RCC. A recently published meta-analysis showed a low objective response rate (about 20%) of this approach in metastatic RCC, with an acceptable safety profile [66]. Based on the published data, currently, rechallenge with immunotherapy should not be considered a standard option in patients with advanced RCC who have progressed on previous CPI-based treatment. In particular, salvage dual checkpoint blockade does not seem to offer good data of activity, whereas a strategy involving the combination of TKI and CPI could be an interesting option. Further data from large randomized clinical trials are needed to better define the role of ICI rechallenge in advanced RCC.

## Figures and Tables

**Table 1 cancers-15-03172-t001:** Summary of selected prospective trials of ICI rechallenge in advanced RCC.

Clinical Trial Identifier	Histology	Line	Phase	Treatment Arm(s)	Accrual	Primary Endpoint(s)	Results
NCT02996110 FRACTION-RCC (track 2)	ccRCC	≥2	II	Nivolumab + Ipilimumab (in ICI-pretreated)	46	ORR, DOR, PFS rate at 24 weeks	ORR 17.4%, DCR 58.7% mDOR 16.4 months PFS rate at 24 weeks 43.2% mPFS 3.7 months mOS 23.8 months Grade 3–4 TRAEs 28.3%
NCT03203473 OMNIVORE (arm B)	ccRCC or nccRCC	≥2	II (adaptive design)	Nivolumab + Ipilimumab (two doses of Ipilimumab for non-responders to Nivolumab induction)	57	Proportion of non-responders to Nivolumab induction converted to response	ORR 4% DCR 50% mPFS 4.7 months OS rate at 18 months 79% Grade 3–4 TRAEs 25%
NCT03117309 HCRN GU16-260 (part B)	ccRCC or nccRCC	2	II (adaptive design)	Nivolumab + Ipilimumab (for non-responders to Nivolumab 1st line therapy)	35	ORR	ORR 11.4% ORR by irRECIST 17.1% Grade 3–4 TRAEs 42.9%
NCT02917772 TITAN-RCC (second-line subgroup)	ccRCC (only intermediate/poor risk by IMDC score)	≥2	II (adaptive design)	Nivolumab + Ipilimumab (2–4 doses of Ipilimumab for non-responders to Nivolumab induction)	139	ORR	ORR 17% DCR 38.9%
NCT01984242 IMmotion150 (second-line part)	ccRCC or sRCC	2	II	Atezolizumab + Bevacizumab (in previously treated with first-line Atezolizumab monotherapy)	44	ORR, PFS, DOR (secondary endpoints of the whole trial)	ORR 25% mPFS 11.1 months
NCT02501096 KEYNOTE-146 (RCC cohort)	ccRCC	≥2	II	Lenvatinib + Pembrolizumab (in ICI-pretreated)	104	ORR at week 24 by irRECIST	ORR at week 24 55.8% ORR 62.5% mDOR 12.5 months mPFS 12.2 months Grade 3–4 TRAEs 64%

Abbreviations: NCT: number of clinical trial (https://clinicaltrials.gov/ accessed on 1 April 2023); ccRCC: clear cell renal cell carcinoma; nccRCC: non-clear cell renal cell carcinoma; sRCC: sarcomatoid renal cell carcinoma; TRAEs: treatment-related adverse events; MTD: maximum tolerated dose; DLT: dose-limiting toxicity.

**Table 2 cancers-15-03172-t002:** Selected ongoing trials of ICI rechallenge in advanced RCC.

Clinical Trial Identifier	Histology	Line	Phase	Treatment Arm(s)	Primary Endpoint(s)	Recruitment Status	Target Accrual
NCT04904302 SNAPI	ccRCC	2–3 (ICI in immediately preceding line)	II	Sitravatinib + Nivolumab	ORR and DCR at 24 weeks	Recruiting	88
NCT04987203 TiNivo-2	ccRCC	2–3	III	Tivozanib + Nivolumab vs. Tivozanib	PFS	Recruiting	326
NCT04338269 CONTACT-03	ccRCC or nccRCC	2–3 (ICI in immediately preceding line)	III	Cabozantinib + Atezolizumab vs. Cabozantinib	PFS, OS	Active, not recruiting	523
NCT03552380 HCRN GU17-326	ccRCC or nccRCC	2 (previously treated with Nivo/Ipi)	II	Entinostat + Nivolumab + Ipilimumab	RP2D (safety lead-in), ORR	Active, not recruiting	18

Abbreviations: NCT: number of clinical trial (https://clinicaltrials.gov/ accessed on 1 April 2023); ccRCC: clear cell renal cell carcinoma; nccRCC: non-clear cell renal cell carcinoma.

## Data Availability

Not applicable.

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
