# Peer review of "Immune Checkpoint Inhibitor Rechallenge in Renal Cell Carcinoma: Current Evidence and Future Directions"

_cancers, 2023, doi:10.3390/cancers15123172_

Round 1

Reviewer 1 Report

Congratulations to the authors. The manuscript represents a well-written review of the immune checkpoint inhibitor rechallenge which is a very interesting and current topic.The article is well structured and fully comprehensive.

Author Response

We are really grateful to the reviewers for their kind appreciation of our paper. We are pleased to have provided a clear overview on this specific topic. We thank again for the comment.

Reviewer 2 Report

The authors provide an overview of the status of immune checkpoint inhibitors in renal cell carcinoma. The comments are included below:

1.    The title of the article is ‘Immune checkpoint inhibitor rechallenge”. However, the authors have not defined what “rechallenge’ is and a distinction between “combination therapy” and “rechallenge” is not made.

2.       A separate “Section 2 under the heading ‘Literature Search’ or ‘Methodology’ or something to that effect should be included to describe the methodology used for literature search. This section could include the source of the literature search (for example PubMed, Clinical trials etc) the timeline, how the authors distinguished combination therapy from rechallenge etc (despite what the cited reference say-critical evaluation of the available literature is a key aspect of review articles). This obviously would include for the authors to decide these studies indeed are rechallenge or combination therapy. Perhaps this distinction could help authors further support their conclusion that “ICI rechallenge should not be considered a standard option (line 392).

3.        What is presented in table 1 is not a ‘schematic summary” (line 294). Also, it is customary to include the date of access.

4.       The failed first line therapy under “Treatment arms” is missing in rows 1 and 3.

Author Response

We would like to thank the reviewers for the effort to review our manuscript. We sincerely appreciate your valuable comments which can help us in improving the quality of our paper.

Point 1. The title of the article is ‘Immune checkpoint inhibitor rechallenge”. However, the authors have not defined what “rechallenge’ is and a distinction between “combination therapy” and “rechallenge” is not made.

Response 1. We appreciated your comment and we have modified our paper by adding a clarification of what we mean for “rechallenge” (lines 59-60) and “combination therapy” (lines 66-67).

Point 2. A separate “Section 2 under the heading ‘Literature Search’ or ‘Methodology’ or something to that effect should be included to describe the methodology used for literature search. This section could include the source of the literature search (for example PubMed, Clinical trials etc) the timeline, how the authors distinguished combination therapy from rechallenge etc (despite what the cited reference say-critical evaluation of the available literature is a key aspect of review articles). This obviously would include for the authors to decide these studies indeed are rechallenge or combination therapy. Perhaps this distinction could help authors further support their conclusion that “ICI rechallenge should not be considered a standard option (line 392).

Response 2. We found your comment very useful in order to enrich our work. So, we added a separate chapter entitled “Literature search and selection of trials” to briefly describe the methodology used in the work.

Point 3. What is presented in table 1 is not a ‘schematic summary” (line 294). Also, it is customary to include the date of access.

Response 3. We accept your suggestion and have modified our manuscript by removing “schematic summary” (line 303). We also reported the date of access (line 303).

Point 4. The failed first line therapy under “Treatment arms” is missing in rows 1 and 3.

Response 4. Thank you for your suggestion. We have specified the failed previous line therapy in rows 1 and 3.